# Development of Gluten-Free Rice Flour Noodles That Suit the Tastes of Japanese People

**DOI:** 10.3390/foods11091321

**Published:** 2022-04-30

**Authors:** Kenjiro Sugiyama, Daichi Matsumoto, Yasuhiro Sakai, Tomoro Inui, Chikako Tarukawa, Masaharu Yamada

**Affiliations:** 1School of Advanced Engineering, Kogakuin University, 2665-1 Nakano-machi, Hachioji 192-0015, Japan; bt13171@ns.kogakuin.ac.jp (K.S.); s217071@g.kogakuin.jp (D.M.); 2Alpha Electronic Co., Ltd., 60-2 Iitoyo-Mukaihara, Tenei-mura 962-0512, Japan; sakai.yasuhiro@alpha-e-net.com (Y.S.); inui.tomoro@alpha-e-net.com (T.I.); tarukawa.chikako@alpha-e-net.com (C.T.)

**Keywords:** rice noodles, gluten-free, potato starch

## Abstract

Gluten-free rice flour noodles with a flavor and texture profile preferred by the Japanese people were developed. The rice noodles contained potato starch (PS) as a binder. “Koshihikari” was selected from several candidate varieties based on its pasting properties. Since the Japanese people prefer the chewy texture of wheat flour “Udon” noodles, first, the stress−strain characteristics of “Udon” noodles in Japan were quantified, using a mechanical test. Next, different formulations of rice noodles were prepared by changing the amount of PS blended into the noodles. The mechanical tests on wheat and rice noodles show that rice noodles made from 85% rice flour and 15% PS have a texture similar to that of “Udon” noodles. Brown rice noodles containing roasted brown rice flour were also developed. Since brown rice flour hinders the binding of the dough, it was necessary to increase the amount of PS to increase the binding of roasted brown rice flour. Finally, noodles with 70% white rice flour, 10% brown rice flour, and 20% PS were produced. The gas chromatography−mass spectrometry analysis of the volatile compounds contained in white rice noodles and brown rice noodles identified the volatile compounds characteristic each of type.

## 1. Introduction

Rice is a staple food of the Japanese population. However, according to data released by the Japanese government, rice consumption per capita has fallen to less than half in the last 50 years [1]. This may be because the Japanese people have come to prefer wheat-flour-containing foods, such as bread and noodles, to rice. This shift may hamper food self-sufficiency in Japan.

Increasing the intake of wheat flour foods by Japanese people may increase the number of people with wheat allergies and celiac disease (CD) due to gluten, which is a hydrate of the protein contained in wheat flour [2]. Celiac disease affects approximately 1% of the world’s population [3]. In Japan, Fukunaga et al. reported that the presence of celiac disease in a non-clinical Japanese population was low, at 0.05% [4]. Although the number of patients suffering from this condition in Japan is smaller than that of Western countries at present, several researchers have pointed out the possibility of an increase in its prevalence in Japan in the near future [5,6]. The only effective and safe treatment of CD remains the strict exclusion of gluten for life under a so-called gluten-free diet [7]. Thus, we aimed to develop gluten-free rice noodles.

Many studies have been performed on the development of gluten-free noodles. Martil et al. extracted starch from parboiled rice flour and produced gluten-free pasta via an extrusion cooking process [8]. They reported that the starch was gelatinized by extrusion and then cooled in order to make the starch retrograde, and the pasta exhibited a hard texture. Sabatini et al. developed gluten-free noodles containing cassava starch, rice flour, and cornstarch as the main ingredients [9]. They reported that replacing native corn starch with pregelatinized corn starch and adding 2% guar gum resulted in an “al dente” texture. Jiao et al. developed a gluten-free noodle using a blend of rice flour and pea starch, and they reported that rice noodles developed a harder texture as the amount of pea starch added increased [10]. Purwandari et al. developed three types of gluten-free noodles using breadfruit, konjac, and pumpkin flour [11]. Of these types of flour, breadfruit flour required fermented cassava flour, and pumpkin flour required cassava flour, as the texturing material. On the other hand, when using konjac flour, no texturing material was required. The hardness, adhesiveness, and cooking loss properties of the noodles made from these flours were in general still higher than those in wheat noodles. All the studies mentioned above aimed for a hard texture, which is referred to as “al dente”.

Toyokawa et al. stated that Japanese people prefer a unique texture; that is, when noodles are chewed, they want to feel a softness but also an elastic resistance [12]. In our study, we chose to call this a chewy texture. White salted udon noodle, which is a typical Japanese wheat flour noodle, is eaten in various parts of Japan. In particular, “Sanuki udon”, favored in Kagawa Prefecture, located on Shikoku Island, is the most popular, and there are many papers on its texture [13,14]. According to these works, Sanuki udon has a chewy texture. Several studies have shown that the texture of “Inaniwa udon”, favored in Akita Prefecture, which is located in the north of Honshu island, has a harder texture than that of Sanuki udon [15,16]. Furthermore, the texture of “Goto udon” in Nagasaki Prefecture is said to be harder than that of “Inaniwa udon”. On the other hand, it is said that the texture of “Hakata udon” from Fukuoka Prefecture and “Ise udon” from Mie Prefecture is soft. However, few studies have compared the textures of udon noodles from all over Japan.

In this study, we first investigated the mechanical properties of wheat udon noodles using several kinds of commercial dry udon noodles, and we identified the parameters that are indicators of a chewy texture, which is a texture preferred by the Japanese people. Next, we selected the rice flour variety to be used for making rice noodles via pasting properties, using a rapid visco analyzer (RVA). Noodles made with 100% rice flour without pressure extrusion absorb less water, preventing the noodles from holding their shape. In this study, we used potato starch (PS) as the binder, as it has a higher water-holding capacity than other starches [17,18,19].

One of the selected rice flour varieties was used to make noodles with different ratios of PS. We then compared the mechanical characteristics of the cooked rice flour noodle with those of cooked udon noodles to identify the components of rice flour noodles that most suit the texture preferences of the Japanese people.

We also developed brown noodles containing brown rice flour to increase the variety of noodles available on the market. Although brown rice bran is healthy, it has a unique odor, and it is difficult to use as food. Therefore, roasted brown flour was selected. Brown noodles contain rice flour, roasted brown rice flour, and PS. 

Another factor relating to the quality of noodles is aroma. Much research on the aroma properties of wheat flour noodles has been conducted. Narisawa et al. analyzed the volatile compounds derived from noodles made from several wheat varieties using headspace solid-phase extraction gas chromatography–mass spectrometry (HS–SPE–GC–MS), discussed the differences in aroma characteristics, and identified compounds characteristic of udon noodles [20,21]. Rice flour noodles have a weaker aroma than udon noodles, so most previous works were performed on brown rice and Indica rice, which have a stronger aroma than Japonica rice. Cao et al. analyzed the aroma of white and brown noodles made from Indica rice using solid-phase microextraction (SPME)–GC–MS [22]. Lin et al. analyzed the volatile compounds in Chinese white salt noodles, and they revealed the characteristic aroma using SPME–GC–MS [23].

In this study, with reference to the above works, we analyzed the volatile compounds of the white and brown rice noodles, using HS–SPE–GC–MS, and we identified the volatile compounds that were characteristic of each noodle.

## 2. Materials and Methods

### 2.1. Udon Noodle Samples

Commercial dry noodle products (“Inaniwa udon”, “Sanuki udon”, “Ise udon”, “Hakata udon”, and “Gotoh udon”) were purchased at a supermarket. The dry noodles were cooked for the standard time described on the package.

### 2.2. Selection and Preparation of the Raw Rice Flour Variety

Based on the ease of procurement of rice, six varieties from the Fukushima Prefecture and one variety from Niigata Prefecture were used as candidates (Table 1). In the table below, the amylose content values from the literature are quoted [24,25,26,27,28]. Amylose content refers to the amylose ratio in the starch contained within the rice. Each type of raw rice was purchased in the form of brown rice, and it was milled and ground by consignment manufacturing (Maruko Foods Inc., Marumori, Japan) to obtain rice flour.

For the development of brown rice noodles, brown rice was ground and roasted (140 °C, 10 min) by consignment manufacturing to obtain brown rice flour.

### 2.3. Preparation and Treatment of PS to Be Mixed as a Binder

The PS was purchased from a starch maker (Nishida Denpun Koujou, Ltd., Hokkaido, Japan). It was blended into rice flour (white or brown) in various ratios. Mixed flour was used for noodle making.

### 2.4. Noodle Making

The rice flour noodles (white noodles, brown noodles) used for the test were manufactured on consignment using a commercial scale process (Maruko Foods Inc., Miyagi, Japan). Rice flour (white or brown flour) and PS were mixed in various ratios. Tap water was added at 44% by mass to the flour. After mixing, the dough was passed into a roller 5 times until a smooth and elastic sheet was formed; this was aged in a refrigerator overnight, cut, steamed, cooled, and vacuum-packed.

### 2.5. Cooking Procedure of Rice Noodles Tested

Water (1 L) was boiled, and approximately 20 g of noodles was cooked for different lengths of time. The texture of noodles is affected not only by its composition but also by the cooking duration. Boiling the noodles for a short duration leads to a firm texture, whereas cooking for a longer time results in a softer texture. To ensure that the water content of the cooked noodle samples, which were boiled for different periods, was the same, the contents were measured to determine the relationship between the boiling time and water content. The water content shows an approximately linear relationship with the boiling time in a realistic boiling time range. The boiling time at which the water content of the rice noodles was 70% (wet basis) was determined from the obtained relationship. The water content was determined from the weight of the noodles before drying and that after drying at 130 °C for 2 h, having allowed them to cool in a desiccator for 30 min.

### 2.6. Quantification of Mechanical Properties

A blade (aluminum, thickness 3 mm, rounded tip) simulating a human tooth was attached to a material test device (EZ-TEST LX, Shimadzu Corporation, Kyoto, Japan) and perpendicularly lowered onto a noodle string placed on a horizontal base fixed to the device used to perform the shear test. The lowering rate of the blade was 10 mm/min. The shear test was carried out within 30 min of cooking. Each sample was analyzed six to eight times. Generally, udon noodles have a rectangular cross-section, and in the mechanical tests, the blade penetrated vertically toward the longer side. The length of this longer side was called the “width” of the noodles. The “width” of the cooked noodles was measured using a vernier caliper set on the base prior to each test. The product of the thickness of the blade and the width of the cooked noodles was designated as the apparent bottom area, and the stress value was obtained by dividing the generated force by the apparent bottom area. The distance from the contact point of the blade with the noodles to the base was defined as unity, and the change in this distance was referred to as the strain. The texture of the noodles was evaluated based on the stress–strain curve.

### 2.7. Pasting Properties

The candidate rice varieties were examined based on their pasting properties, which were quantified using the rapid visco analyzer (RVA: RVA4500, PerkinElmer, Waltham, MA, USA).

A sample of rice noodles (3.50 g) with 14% water content was weighed and suspended in 25 mL of water, and then, pasting properties were analyzed using the RVA. The measurement cell was held at 50 °C for 1 min, heated from 50 °C to 93 °C for 4 min, held at 93 °C for 7 min, and then cooled to 50 °C for 4 min, before finally being held at 50 °C for 3 min, based on the procedure of Nakamura et al. [29]. Each sample was analyzed three times. 

### 2.8. Analysis of Volatile Compounds

A quadrupole GC−MS (GC: 7890A, Agilent Technologies, Santa Clara, CA, USA; MS: JMS-Q1000GC Mk-II, JEOL, Tokyo, Japan) was used for the analysis of the volatile compounds present in the noodles. A DB-WAX capillary column (Agilent Technologies, Santa Clara, CA, USA; inner diameter, 0.25 mm; length, 60 m; film thickness, 0.25 µm) was used for gas chromatography, and high-purity He (purity 99.9995%) was used as the carrier gas (flow rate: 1.5 mL/min).

An uncooked noodle sample (3.0 g) was placed in a 20 mL vial, and the headspace gas in the vial, which was thermally equilibrated at 80 °C for 20 min, was absorbed onto a solid phase (GL trap1; GL Science Inc., Tokyo, Japan) by pressurizing the vial at 140 kPa in an autosampler (S-trap; JEOL, Tokyo, Japan). After absorption, the solid phase was dried by flowing He for 1.5 min and then heated at 200 °C for 3 min to desorb and inject gas into the GC column. The GC oven temperature was maintained at 40 °C for 10 min, increased to 220 °C at a rate of 5 °C /min, and maintained at 220 °C for 10 min.

An n-alkane reagent (C8 to C20 in n-Hexane; GL Science Inc., Tokyo, Japan) was used to determine the Kovats’ retention index (RI) [30], and 1 mL of 2-methyl-2-butanol aqueous solution (1 ppm) was used as the external standard sample.

The obtained chromatogram was deconvolved using the automated mass spectral deconvolution and identification system (AMDIS; National Institute of Standards and Technology (NIST), Gaithersburg, MD, USA), and candidate compounds were identified using MS-Search (NIST, Gaithersburg, MD, USA) [31]. For each candidate compound, the RI obtained from the analysis of the n-alkane reagent was calculated, and the compound was identified using the mass spectrum of the pure compound and the RI database provided in the NIST Chemistry WebBook [32]. The target ion was determined when the compound was identified, and the peak area of the target ion was used as a measure of the concentration of each compound in the noodle sample. Each sample was analyzed three times.

### 2.9. Statistical Analysis

Tukey’s HSD test was applied for the comparison of yield stresses between cooked rice noodles using R version 3.1.3 (R Core Team, Vienna, Austria) [33]. Statistical significance was assessed with a confidence level of 0.99 (*p* = 0.01). A principal component analysis (PCA) of the volatile compound data was performed using the 64-bit statistical analysis tool and Excel 2010 (RIKEN Center for Sustainable Resource Science, Saitama, Japan) [34]. The PCA is a statistical method whereby a multidimensional phenomenon is reduced to a few dimensions. The principal components 1 (PC1) and 2 (PC2) emerge as variables when a multidimensional phenomenon is reduced to two dimensions, and information is lost as a result of this reduction [35]. The “proportion of variance” is a measure of how much information is retained.

## 3. Results and Discussion

### 3.1. Mechanical Properties of Cooked Udon Noodles

The primary aim of the study was to develop rice noodles that suit the taste preferences of the Japanese people, while trying to simultaneously achieve a texture similar to that of udon noodles. Therefore, in order to identify the numerical index of a chewy texture, we examined the texture of udon noodles.

The representative stress–strain characteristics of five kinds of cooked commercial udon noodles are shown in Figure 1. In all cases, the stress increased from the start of the penetration of the blade, before peaking. At this value, noodles break, causing stress relaxation. Consequently, the stress value drops, the blade contacts the base, and the test is completed. Since the peak stress value corresponds to the force exerted when a human tooth bites and breaks a noodle, it can be used as an index of the chewy texture. This stress value is called the yield stress, according to material mechanics. “Gotoh udon” and “Inaniwa udon” noodles show sharp peaks near the yield point; “Sanuki udon” and ”Hakata udon” noodles show upwardly convex stress–strain curves, indicating that plastic deformation occurs during the shear test, which suggests a chewy texture that suits the tastes of the Japanese people; and “Ise udon” noodles show a broad peak. These “Ise udon” noodles have a texture that is too soft for Japanese people, and therefore the noodles are thickened. Furthermore, in the case of “Gotoh udon” and ”Inaniwa udon” noodles, the noodles are thinned because the texture is too firm. Thus, we aimed for a texture in our rice noodles that is the same as that of “Sanuki udon” noodles. These discussions indicate that the texture and the width of the noodles are related to each other. Therefore, we also plotted the relationship between the width of the cooked noodles and the yield stress of the stress–strain curve. The results are shown in Figure 2. The yield stress values correspond to the peak stress values in Figure 1. We found that a specific relationship holds, as shown in Figure 2. It is important to note that this relationship is based on Japanese food preferences, and it does not have scientific criteria. When measuring the width and yield stress of cooked rice noodles, we tried to determine the formulation of rice noodles, aiming for the texture of “Sanuki udon”.

### 3.2. Selection of Raw Rice Flour Variety

A representative RVA measurement result (Koshihikari flour) is shown in Figure 3. The viscosity in the cell increases sharply as the temperature rises before reaching its maximum value (peak viscosity). After that, the viscosity reaches its lowest value (trough viscosity). The viscosity begins to increase as cooling begins, and it finally approaches a constant value (final viscosity). The difference between the peak viscosity and trough viscosity values is called the breakdown value, and it is used as an index of gelatinization. The difference between the trough viscosity and final viscosity values is called the consistency value, which is a measure of how hard the gel becomes upon cooling [36]. In terms of noodle texture, the breakdown value is related to the chewiness of the noodles, and the consistency value is related to the tendency of the noodle to harden after cooking. In this study, we evaluated the difference in the pasting properties of samples using breakdown and consistency values. 

The relationship between the breakdown and consistency values of rice flour samples is shown in Figure 4. Although the “Koshinokaori (Niigata)” variety had the largest breakdown value, it also had a large consistency value. It would not be appropriate to select this variety, because noodles made from this variety tend to harden over time after cooking. The “Kamenoo”, “Koshinokaori (Fukushima)”, and “Tennotubu” varieties were excluded because their breakdown values were relatively small; consequently, their chewy texture would likely be weak. 

The “Sasashigure”, “Hitomebore”, and “Koshihikari” varieties from Fukushima Prefecture were selected as the candidate varieties (circled red dashed line) because of their relatively large breakdown values and low consistency values. We decided to use “Koshihikari”, which has the lowest consistency value among the three candidate varieties; that is, it is resistant to retrogradation, according to various analyses and tests. 

In addition, brown rice flour showed small breakdown (1646 mPa s, SD282) and consistency (1167 mPa s, SD30) values, and PS showed an excessive breakdown value (1.000 × 10^4^ mPa s, SD1448) and a medium consistency value (1777 mPa s, SD472). Here, SD means standard deviation. These data were not plotted in Figure 4 to ensure the readability of the other data. The low breakdown value of brown rice flour results in difficulty when making the dough. The large breakdown value of PS contributes to its chewy texture, but excessive blending induces hardness in the noodles.

### 3.3. Percentage of PS in the Rice Noodles

Gluten-free noodle dough cannot contain 100% rice flour because a binder to replace gluten is required. In this study, PS was chosen as the binder. The higher the amount of PS, the easier it is to form noodle dough; however, the texture becomes firm beyond chewy. On the contrary, if the content is too low, the dough cannot be appropriately prepared, and the noodles collapse during the noodle-making process. Therefore, the optimum amount of PS should be determined based on the mechanical properties of cooked noodles.

Different formulations of rice noodles were made by changing the amount of PS added to the noodle dough, i.e., from 5% to 20%. A PS addition rate of 5% indicates that the flour contains 5% PS on a mass basis. At 5% PS, the dough cannot be formed. We then decided to increase the PS addition rates to 10, 15 and 20%. 

On the other hand, in the case of brown noodles, the addition of roasted brown flour impairs the ability to form a dough, and so the addition was therefore limited to 10%. Therefore, we set the blend amount of roasted brown rice powder to 10% and conducted a test in which the amount of PS added was changed to adjust the texture. When the PS was set to 15%, the noodles remained brittle, and when the amount of PS added was 20%, the brittleness was improved. Even then, the width of the noodles needed to be 7 mm rather than 4 mm.

Table 2 shows the formulations and widths of the cooked rice noodles examined in this study. RUN1 to 4 show white rice noodles, and RUN5 shows a brown rice noodle. A representative stress–strain curve of each condition is shown in Figure 5. The yield stress value tended to decrease as the amount of PS added decreased. 

A bar graph plotted using all the yield stresses of these five formulations is shown in Figure 6. There was no significant difference at 20% PS, even if the width of the noodles changed. However, the yield stress decreased significantly with a decrease in the blending ratio of PS. This graph also shows the data for “Sanuki udon”. It can be seen that the values of “Sanuki udon” are similar to those of the 15% PS-supplemented rice flour noodle.

The measurement results of the yield stress and widths of the rice flour noodles are superimposed on the relationship diagram of the yield stress and width of the udon noodles. The results are shown in Figure 7. The data points of udon noodles are indicated by black circle symbols, “●”. The data on rice flour noodles intersect with those of udon noodles, and the data of the 15% PS-supplemented noodles just overlap those of udon noodles. The data of the udon noodles in the overlapping part correspond to those of “Sanuki udon”. This means that the texture of rice flour noodles with the addition of 15% PS is similar to that of “Sanuki udon”.

### 3.4. Aroma Characteristics

We analyzed the aroma characteristics of the white and brown rice noodles before cooking. The GC–MS analyses detected 39 compounds in common between white rice noodles and brown rice noodles. Table 3 shows a list of the detected compounds, with the corresponding RI and target ion. The values obtained from the analysis are designated as RI_exp, and the values from the database are designated RI_ref. In Table 3, the unknown compounds refer to those for which there were no polar column data in the database. The peak areas of the target ions were determined for these components. 

The results of the PCA are shown in Figure 8; (A) represents a score plot, and (B) represents a loading plot. In this analysis, the proportion of variance of the first principal component (PC1) was 69%, and that of the second principal component (PC2) was 19%. This means our analysis represents 88% of the original information.

The score plot shows that both samples were mapped to different positions. The loading plot shows that the positions of the white rice noodles were determined by esters, such as ethyl acetate and ethyl 2-hydroxyhexanoate, alcohols, such as 2-ethyl-1-hexanol and 1-octanol, and aldehydes, such as nonanal and (E)-2-nonenal. In particular, alcohols and aldehydes have been shown to contribute to the aroma of rice [37]. On the other hand, the positions of brown rice noodles were determined by their low boiling point, as well as their content of sweet-flavored aldehydes such as 2-methyl-propanal, 2-methyl-butanal and 3-methyl-butanal, Maillard reaction products such as furfural, and nut-flavored compounds such as benzaldehyde and acetophenone [38]. The Japanese people have historically tried to incorporate roasted aromas into their traditional foods by applying a Maillard reaction, utilizing ingredients such as miso and soy sauce. The white rice noodles developed in this study have a rich rice flavor, whereas the brown rice noodles have a roasted aroma, which is preferred by the Japanese people.

In addition, the signals of the volatile compounds in the cooked rice noodles were weak, and it was hard to sufficiently compare the white rice noodles with the brown rice noodles. Therefore, in this study, we analyzed the aroma characteristics of the noodles before cooking. The analysis of the aroma of the noodles after cooking is an issue to be addressed in the future, with the examination of analysis conditions. Furthermore, the scientific sensory evaluation of aroma could not be carried out due to the limitations of human resources. Sensory tests will be performed in further studies.

## 4. Conclusions

In this study, we aimed to develop gluten-free rice noodles with a texture and flavor profile preferred by the Japanese people. The most appropriate raw rice flour for the noodles was selected based on its pasting properties. PS was used as the binder, and the blending ratio was set to 15% based on the stress–strain characteristics of the cooked noodles and upon comparison with those of udon noodles. In addition to the white rice noodles, as a product variation, brown rice noodles containing roasted brown rice flour were also developed. Both white and brown rice noodles were evaluated for their aroma, and the two types exhibited different aroma characteristics. We are currently offering prototypes of our developed product to the market, and they have been well-received. We believe that these products will enhance the gluten-free food industry.

Issues to be addressed in the future include scientific sensory evaluations of the texture and aroma characteristics, GC–MS analyses of the cooked rice noodles, and the development of brown rice noodles with better texture and flavor. These challenges will contribute to further improving the quality of gluten-free rice flour noodles.

## Figures and Tables

**Figure 1 foods-11-01321-f001:**
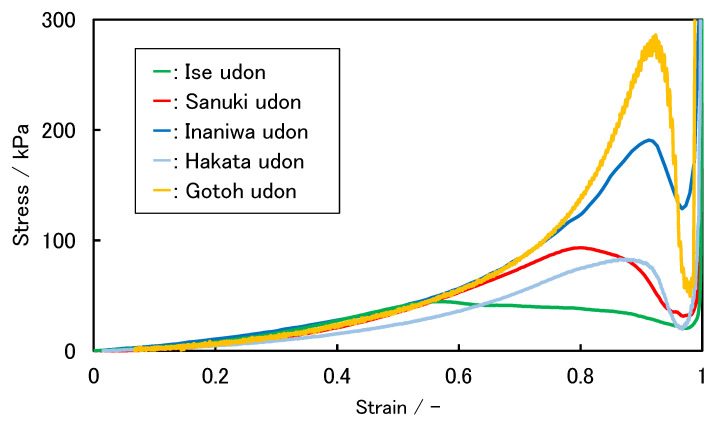
Representative stress–strain curves of various udon noodles: a representative measurement result of each condition.

**Figure 2 foods-11-01321-f002:**
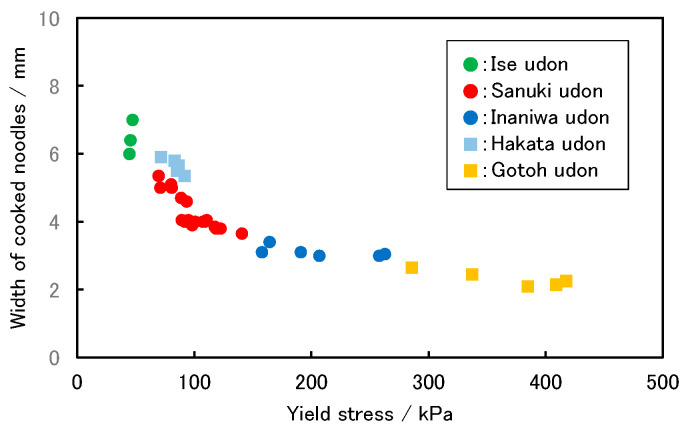
Relationship between the yield stress and width of udon noodles.

**Figure 3 foods-11-01321-f003:**
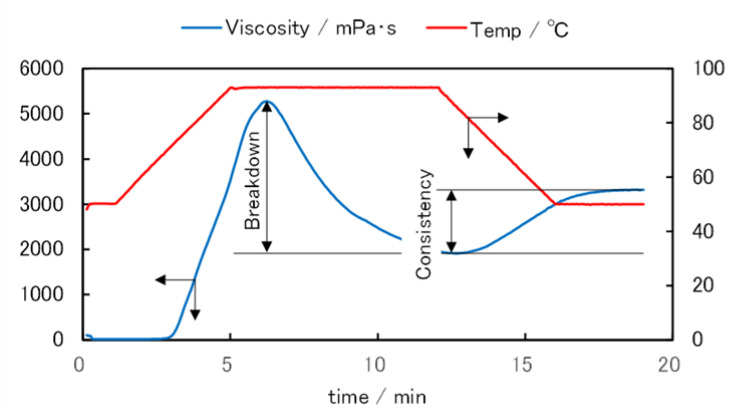
A representative measurement result (Koshihikari) and a definition of breakdown and consistency.

**Figure 4 foods-11-01321-f004:**
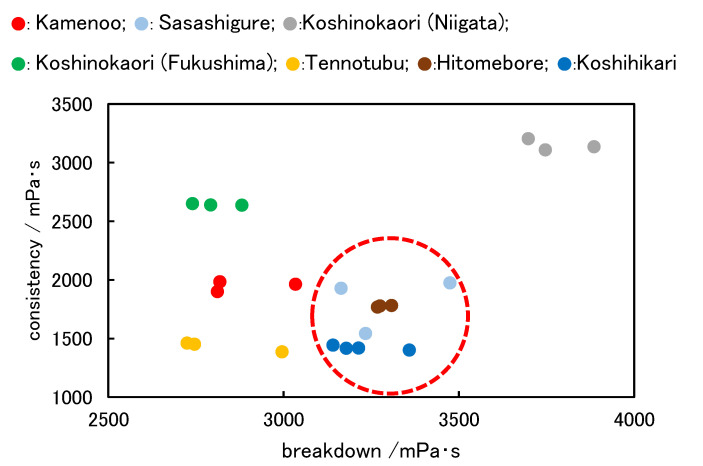
Relationship between the breakdown and consistency of raw rice candidates.

**Figure 5 foods-11-01321-f005:**
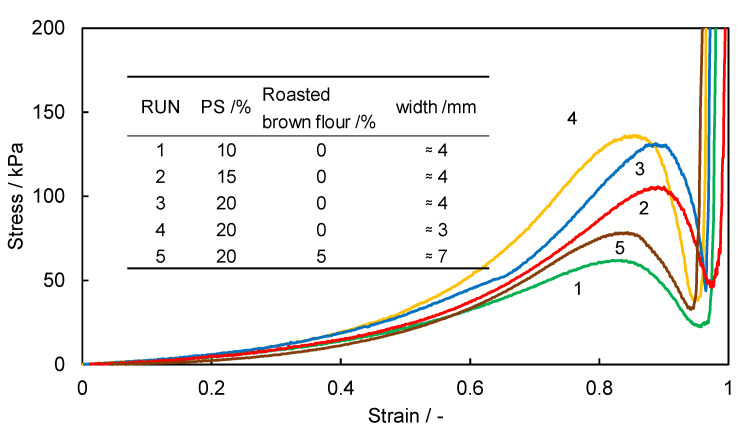
Stress–strain diagram of cooked rice noodles: a representative measurement of each condition. “≈” means “approximately equal”.

**Figure 6 foods-11-01321-f006:**
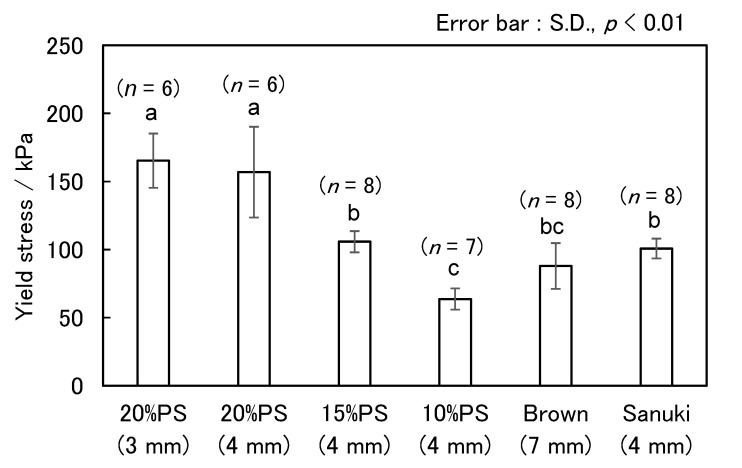
Comparison of yield stresses between cooked rice noodles. The length in parentheses on the horizontal axis indicates the width of the noodles. The “*n*” above the bar represents the number of measurements for each condition. The error bar shows the standard deviation (SD). Bars with different letters indicate significant differences (*p* < 0.01, Tukey’s test).

**Figure 7 foods-11-01321-f007:**
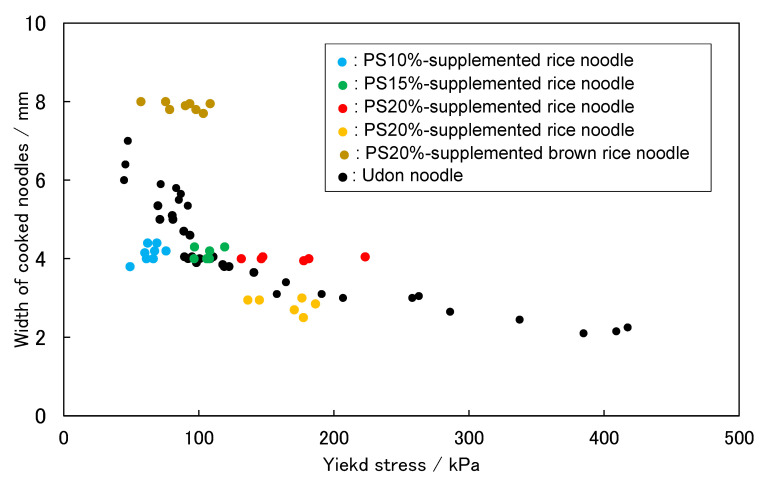
Relationship between yield stress and width of cooked noodles.

**Figure 8 foods-11-01321-f008:**
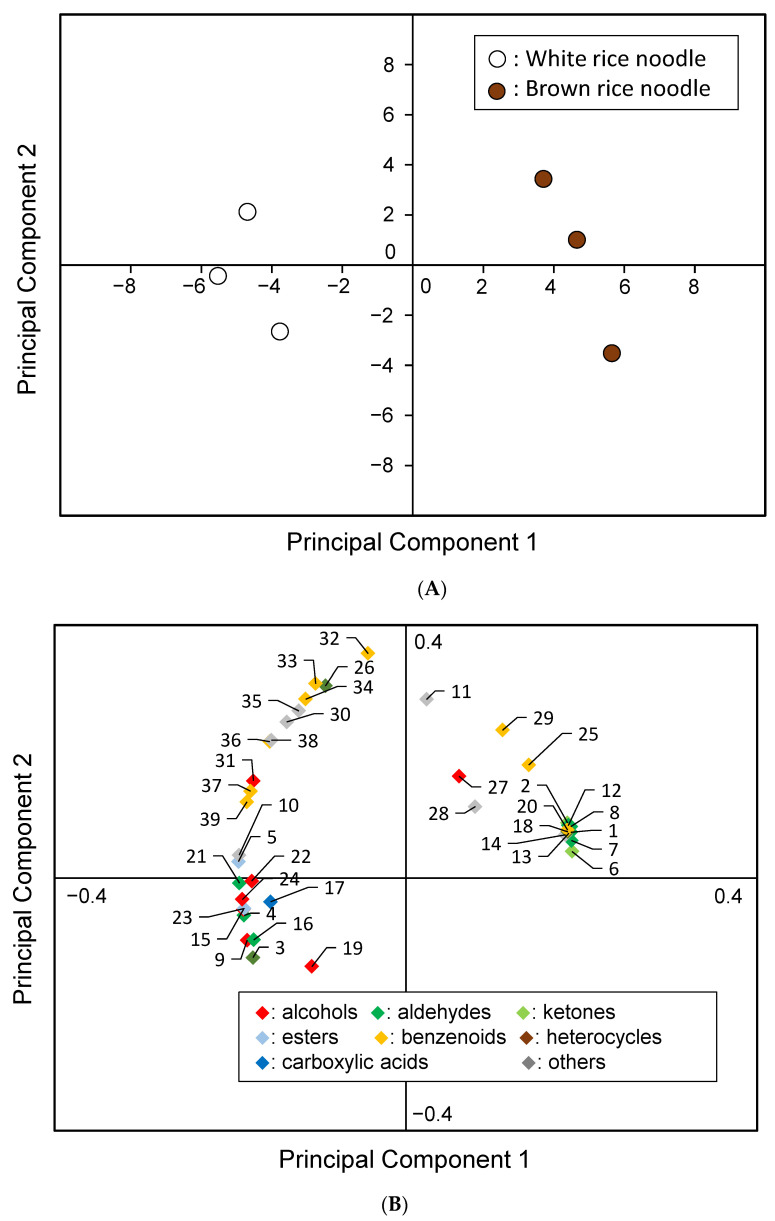
Results of PCA. The numbers represent the values in Table 3. (**A**) Score plot. White circles represent white rice noodles, brown circles represent brown rice noodles. (**B**) Loading plot.

**Table 1 foods-11-01321-t001:** Candidate rice varieties.

No.	Variety	Area	Amylose Content/%	Reference
1	Kamenoo	Fukushima	16.5	[24]
2	Sasashigure	Fukushima	19.4	[25]
3	Koshinokaori	Niigata	33.7	[26]
4	Koshinokaori	Fukushima	33.7	[26]
5	Ten-no-tsubu	Fukushima	19.4	[27]
6	Hitomebore	Fukushima	18.5	[28]
7	Koshihikari	Fukushima	15.6	[24]

**Table 2 foods-11-01321-t002:** Formulation and width of the cooked rice noodles examined in this study.

RUN	Rice Flour/g	Roasted Brown Flour/g	PS/g	Water/g	Width/mm
1	90	0	10	44	≈4
2	85	0	15	44	≈4
3	80	0	20	44	≈4
4	80	0	20	44	≈3
5	70	10	20	44	≈7

“≈” means “approximately equal”.

**Table 3 foods-11-01321-t003:** Volatile compounds in white and brown rice noodles.

No.	Compound	RI_exp	RI_ref	Target Ion
1	Propanal, 2-methyl-	812	801	72
2	Acetone	815	813	43
3	Furan, tetrahydro-	838	861	42
4	Butanal	843	877	72
5	Ethyl acetate	852	884	43
6	2-Butanone	862	893	43
7	Butanal, 2-methyl-	871	907	57
8	Butanal, 3-methyl-	875	911	44
9	Ethanol	915	933	45
10	Trichloromethane	1017	1013	83
11	Unknown	1042	-	75
12	Hexanal	1102	1097	56
13	1-Pentanol	1322	1255	55
14	Unknown	1392	-	57
15	1-Hexanol	1399	1354	56
16	Nonanal	1412	1390	57
17	Acetic acid	1476	1433	60
18	Furfural	1480	1466	96
19	1-Hexanol, 2-ethyl-	1495	1491	57
20	Benzaldehyde	1524	1520	77
21	(E)-2-Nonenal	1544	1532	70
22	1-Heptanol, 6-methyl-	1577	1524	55
23	Ethyl dl-2-hydroxycaproate	1582	1552	69
24	1-Octanol	1590	1565	69
25	Acetophenone	1612	1632	77
26	3-Furanol, tetrahydro-	1622	1619	57
27	2-Octen-1-ol, (E)-	1631	1605	57
28	Unknown	1677	-	118
29	Benzene, octyl-	1718	1721	91
30	1-Hexene, 2,5,5-trimethyl-	1722	no data	57
31	3-Octen-2-ol	1746	1769	71
32	Benzene, (1-propylheptyl)-	1773	1743	91
33	Benzene, (1-ethyloctyl)-	1778	1767	91
34	Benzenepropanal	1788	1783	91
35	Unknown	1806	-	91
36	Benzene, (1-pentylhexyl)-	1830	1820	91
37	Benzene, (1-butylheptyl)-	1835	1828	91
38	Unknown	1843	-	105
39	Benzene, (1-propyloctyl)-	1847	1843	91

The values obtained from the analysis are designated as RI_exp, and those from the database are designated RI_ref. “-” means that an RI value cannot be described because the compound name is not decided.

## Data Availability

The data presented in this study are available on request from the corresponding author.

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
