# Peer review of "Development of Gluten-Free Rice Flour Noodles That Suit the Tastes of Japanese People"

_foods, 2022, doi:10.3390/foods11091321_

Round 1

Reviewer 1 Report

Abstract section should be improved. Inhibiting the use of we and our.

Introduction section: more references should be cited. Some paragraph should be merged. More background should be introduced.

Statistical analysis method should be added.

The discussion is not adequate. Also, more references should be cited. 

The English of the whole manuscript should be improved.

Author Response

First of all, thank you for your peer review. Our answers to the questions and comments are shown in the attachment.

Reviewer 2 Report

The manuscript entitled “Development of gluten-free rice flour noodles that suit the tastes of Japanese people” presents interesting issue, however some corrections are needed. 

  • Abstract – some detailed data (findings) should be presented.
  • Lines 40-41 – ‘The Japanese people prefer a unique texture profile for noodles; they prefer noodles 40 that have a chewy texture.” – please add some references
  • Lines 42-44 – “Sanuki udon” noodles are chewy; “Gotoh udon” and “Inaniwa udon” noodles are firm, while noodles like “Hakata udon” and “Ise udon” are soft and chewy” – international readers may be not familiar with this type of udon – please specify it.
  • In introduction section Authors presented the information associated with the texture of noodle. This section authors should presents – what do we know and what is the background for this study. Some detailed information about other studies are necessary (international context – the data/ studies from other countries should be presented). The good background should present the history of problem, the current knowledge and scientific "gap", and then authors should present how their study could fill this gap to justify the study.
  • The clear aim of the study should be presented at the end of the introduction.
  • Line 361 – ‘Each sample was analyzed three times.’ This sentence belongs to the material and method section.
  • Discussion section - Authors should in their discussion include 3 areas: (1) compare gathered data with the results by other authors, (2) formulate implications of the results of their study and studies by other authors, (3) formulate the future areas which should be studied. Authors should present here and discuss the limitations of their study.
  • The study is interesting but some information is missing. The intercultural contexts of should be presented – due to the fact that for international regardless the Japanese preferences for chewing texture may be unknown. Moreover, sensory analysis should be conducted to verify the taste, texture and flavor of noodle (if such data are not available – please indicate it as a limitation of the study).   

Minor comments:

  • Some typos occurs – e.g. line 145 ‘[13], For” – it should be dot instead of coma

Author Response

(The authors gave the same response as above.)

Reviewer 3 Report

The manuscript entitled „Development of gluten-free rice flour noodles that suit the tastes of Japanese people” investigated the possibility of producing a gluten-free version of wheat-flour “Udon” noodles, using a combination of rice flour and potato starch. Therefore, the authors first determined the stress-strain characteristics of wheat flour noodles and selected an appropriate rice flour based on he pasting properties. Then rice flour/potato starch noodles were produced aiming to obtain a consistency similar to the wheat noodles. The aroma of the noodles was evaluated by determination of volatile compounds using GC-MS.

The scientific design of the study appears sound but there are far too less details given in the Materials and Methods section. The results are presented clearly but not discussed sufficiently. The English is mostly fine. The subject of gluten-free foods is of high interest, due to the high number of people suffering from celiac disease. However, the product investigated is a local food. Therefore, I`m wondering about the market relevance these rice noodles could have outside of Japan. The authors should try to provide a more global field of application of their results.

Comments:

Line 15: omit “other” because potato starch is not a grain starch

Line 37: please use the scientific term for gluten intolerance, which is “celiac disease”, please cite a reference for the annually increasing number

Line 43 and throughout the manuscript: Please remove the “;” and rephrase the sentence accordingly.

Line 51: potatoes are no grains, therefore you cannot state “we used potato starch (PS) as the binder, as it has a higher water-holding capacity than other grain starches.” Please correct.

Line 53: omit words “what” and “the”

Line 30 – 62: The entire Introduction contains only one reference. Statements, such as potato starch has high water binding or rice flour doesn`t bind enough water for the noodles to hold their shape are only examples for statements in the Introduction, that require a reference. Please add appropriate references to all statements necessary. Also, I`m wondering if there is a market relevance for “chewy noodles” outside of Japan, please explain. Gluten and celiac disease are stated but never explained. What is gluten, what is celiac disease, what are gluten-free raw materials and what are common approaches to produce gluten-free products. Have there been studies on gluten-free Japanese noodles? If so, what are the knowledge gaps that this study is aiming to fill? Are there studies that used rice flour or potato starch to produce gluten-free products, maybe even noodles? If so, you should cite them and explain the differences to your study, if not you should state what others used instead and why you decided for rice and potato.
What about the brown rice noodles you only state in one sentence? They should be introduced properly as well. What about the volatile compounds of wheat noodles? Are they at least partially known? Then you should write something about them in the introduction. If they are not known, what does the aroma of the gluten-free noodles tell you? Please explain.
These are some issues showing that the introduction is insufficient in its current form and needs extensive revision.

Line 68/69: Please state the model and supplier of the RVA and explain the method used to determine the pasting properties.

Table 1: Please add a column with the appropriate references for each amylose content. Is the amylose content based on the whole flour or the starch?

Line 74-80: This explanation is more suitable for the introduction or the discussion of your results. At this point in the manuscript you should state the different levels of PS you added to the rice flour and how you determined the ideal level of PS in the mixture. Please revise.

Line 83-93: This paragraph contains some introduction on why you examined the cooking time and some sort of discussion of the results obtained. However, both aspects are misplaced in the Materials and Methods section. Instead, you should describe how you treated your samples and how you determined different parameters. In this section it is not sufficient to state that you tried different boiling times, instead you need to state the different times.

Line 110: Please check, did you use rice noodles or rice flour for this measurement?

Line 110-122: Please give more information on your measurements. For example, how much water did you add to the 3.5 g of sample? How many replicate measurements did you carry out? Was the temperature profile based on a reference?

Figure 1: Did you obtain this curve by one of your own measurements? If so, please state what kind of sample was measured. If not, please cite a reference from where you took the curve.

Line 133: Why was an uncooked noodle sample used? Noodles are consumed after cooking, so the profile of volatile compounds after cooking is the important factor for consumer acceptability.

Materials and Methods: How did you prepare the noodles? Please provide the recipe and details of the methodology used to process the rice flour into cooked noodles. What about statistical analysis? Did you do any replicate measurements and any statistics to verify, that differences are significant? If so, please state in this section what you did. Also, information on the potato starch used are missing completely.

Also, I`m a little confused. Major axis = diameter? Please explain the expression “Major axis”.

Figure 2: since all 5 types of commercial wheat noodles show different stress-strain curves I`m a little confused which behaviour was used as “goal” for the rice flour/potato starch noodles. Please explain.

Figure 3: You state that there is a specific relationship plotted in this figure. But what are you doing with this information? I`m wondering again, which of the five commercial noodles will be used as reference.

Line 189/190: Why is a high consistency bad for making noodles, please explain.

Line 188-194: Seven varieties were studied, but only 4 are mentioned in the Text, what about the other three? Out of the 4 varieties mentioned, only the results of one are slightly discussed. Please discuss your results more detailed. Out of the three candidate-varieties, why did you choose Koshihikari?

Line 211: “A typical stress-strain curve…” What do you mean by typical. A representative measurement of your samples or a curve you took from a reference that is typical for flours containing potato starch?

Line 217-219: This information would be more suitable for the figure title than the main text body.

Line 217: Why 6-8 measurements? Why is the number of measurements not equal for all samples? Like this you need to indicate which samples had n=6 and which had n=8.

Line 221/222: The yield stress of which PS ratio corresponds with the yield stress of what commercial wheat flour noodle. Please explain in more detail.

Line 231: Why was there no sensory evaluation mentioned in the manuscript so far? This should be an important chapter in the Materials and Methods section. Why are there no sensory-scores presented to show the sensory evaluation of the noodles? In a study aiming at consumer preference, this should be an elemental part of the work and be described in great detail. Also, I`m wondering if the texture measurements are sufficient to evaluate sensory and texture properties of the noodles. How do the results compare to the sensory evaluation?

Line 239: Why is the development of the brown rice noodles not explained in the Materials and Methods section? Where did you obtain the brown rice flour? How are the pasting properties compared to white rice flour or wheat flour? Did you roast it yourself? If so, please explain why?

Line 246: What studies, if you mention them in the text you also have to cite them.

Line 240-249: This whole chapter about brown rice noodles appears out of nowhere. It is not introduced in the introduction, the methodology is not explained or even mentioned in the Materials and Methods and no real results are shown and discussed in this section. In its current form, the paragraph has no benefit for the manuscript and confuses the reader.

Line 252: Which ones, the white ones or the brown ones?

Line 257: 17 identified components are a lot less than 39, so how are the results comparable?

Line 252-262: What about the individual compounds? Did you identify the same compounds as the references you cited? Where are the differences in the compounds you identified and your references identified? Dis you also find more aroma compounds in brown noodles than in white ones? Are the aroma compounds found in agreement with the sensory evaluation of the cooked noodles? Please discuss these questions. In Materials and Methods you stated, that you also quantified the volatile compounds, please show and discuss these results as well.

Figure 8: What are the principle components 1 and 2? Please explain all details of the PCA in the materials and Methods section.

Line 295-304: Are there any future research questions, that arise from your work? Please discuss.

Author Response

First of all, thank you for your detailed peer review. Our answers to the questions and comments are shown in the attachment.

Round 2

Reviewer 1 Report

The authors addressed our suggestion. I have no more suggestions.

Author Response

Thank you for your peer review again. Finally, I have asked the MDPI Language Editing Service to resubmit our revisions. Thank you again for your cooperation.

Reviewer 2 Report

I appreciate the great efforts that the authors have made in response to my questions and concerns. However, there are some issues that should be corrected:

  • no need for bolding (lines 206-211)
  • in statistical analysis - The information about α (p-Value) is needed (level of significance)

Author Response

(1) Thank you for your advice. I made mistake.

      I modified the part to the standard font.

(2) Thank you for your suggestion.

      In “2.9. Statistical analysis”, I added the following sentence.

Statistical significance was assessed with a confidence level of 0.99 (p= 0.01).

Reviewer 3 Report

The authors have addressed and resolved all issues raised by me regarding the manuscript. The manuscript has undergone extensive revisions and is much improved now. 

Author Response

(The authors gave the same response as above.)
